# Treatment of C3 Glomerulopathy in Adult Kidney Transplant Recipients: A Systematic Review

**DOI:** 10.3390/medsci8040044

**Published:** 2020-10-21

**Authors:** Maria L Gonzalez Suarez, Charat Thongprayoon, Panupong Hansrivijit, Karthik Kovvuru, Swetha R Kanduri, Narothama R Aeddula, Aleksandra I Pivovarova, Api Chewcharat, Tarun Bathini, Michael A Mao, Arpita Basu, Wisit Cheungpasitporn

**Affiliations:** 1Division of Nephrology, Department of Internal Medicine, University of Mississippi Medical Center, Jackson, MS 39216, USA; apivovarova@umc.edu; 2Division of Nephrology and Hypertension, Department of Medicine, Mayo Clinic, Rochester, MN 55905, USA; 3Department of Internal Medicine, University of Pittsburgh Medical Center Pinnacle, Harrisburg, PA 17101, USA; hansrivijitp@upmc.edu; 4Department of Medicine, Ochsner Medical Center, New Orleans, LA 70121, USA; karthik.kovvuru@ochsner.org (K.K.); svetarani@gmail.com (S.R.K.); 5Division of Nephrology, Department of Medicine, Deaconess Health System, Evansville, IN 47710, USA; dr.anreddy@gmail.com; 6Department of Medicine, Mount Auburn Hospital, Harvard Medical School, Cambridge, MA 02138, USA; api.che@hotmail.com; 7Department of Internal Medicine, University of Arizona, Tucson, AZ 85721, USA; tarunjacobb@gmail.com; 8Division of Nephrology and Hypertension, Department of Medicine, Mayo Clinic, Jacksonville, FL 32224, USA; mao.michael@mayo.edu; 9Emory Transplant Center and Department of Medicine, Renal Division, Emory University School of Medicine, Atlanta, GA 30322, USA; arpita.basu@emory.edu

**Keywords:** C3 glomerulopathy, C3 glomerulonephritis, dense deposit disease, kidney transplantation, renal transplantation

## Abstract

Background: C3 glomerulopathy (C3G), a rare glomerular disease mediated by alternative complement pathway dysregulation, is associated with a high rate of recurrence and graft loss after kidney transplantation (KTx). We aimed to assess the efficacy of different treatments for C3G recurrence after KTx. Methods: Databases (MEDLINE, EMBASE, and Cochrane Database) were searched from inception through 3 May, 2019. Studies were included that reported outcomes of adult KTx recipients with C3G. Effect estimates from individual studies were combined using the random-effects, generic inverse variance method of DerSimonian and Laird., The protocol for this meta-analysis is registered with PROSPERO (no. CRD42019125718). Results: Twelve studies (7 cohort studies and 5 case series) consisting of 122 KTx patients with C3G (73 C3 glomerulonephritis (C3GN) and 49 dense deposit disease (DDD)) were included. The pooled estimated rates of allograft loss among KTx patients with C3G were 33% (95% CI: 12–57%) after eculizumab, 42% (95% CI: 2–89%) after therapeutic plasma exchange (TPE), and 81% (95% CI: 50–100%) after rituximab. Subgroup analysis based on type of C3G was performed. Pooled estimated rates of allograft loss in C3GN KTx patients were 22% (95% CI: 5–46%) after eculizumab, 56% (95% CI: 6–100%) after TPE, and 70% (95% CI: 24–100%) after rituximab. Pooled estimated rates of allograft loss in DDD KTx patients were 53% (95% CI: 0–100%) after eculizumab. Data on allograft loss in DDD after TPE (1 case series, 0/2 (0%) allograft loss at 6 months) and rituximab (1 cohort, 3/3 (100%) allograft loss) were limited. Among 66 patients (38 C3GN, 28 DDD) who received no treatment (due to stable allograft function at presentation and/or clinical judgment of physicians), pooled estimated rates of allograft loss were 32% (95% CI: 7–64%) and 53% (95% CI: 28–77%) for C3GN and DDD, respectively. Among treated C3G patients, data on soluble membrane attack complex of complement (sMAC) were limited to patients treated with eculizumab (N = 7). 80% of patients with elevated sMAC before eculizumab responded to treatment. In addition, all patients who responded to eculizumab had normal sMAC levels after post-eculizumab. Conclusions: Our study suggests that the lowest incidence of allograft loss (33%) among KTX patients with C3G are those treated with eculizumab. Among those who received no treatment for C3G due to stable allograft function, there is a high incidence of allograft loss of 32% in C3GN and 53% in DDD. sMAC level may help to select good responders to eculizumab.

## 1. Introduction

C3 glomerulopathy (C3G) is a rare glomerulonephritis (GN) characterized by the dysregulation of the alternative complement pathway in the glomeruli. It leads to prominent complement C3 deposition in the renal biopsy samples with absent or scanty immunoglobulin deposition [1,2,3,4,5,6,7,8,9,10,11,12]. Microscopically, C3G may present as patterns of membranoproliferative GN (MPGN), crescentic GN, diffuse endocapillary proliferative GN, or mesangioproliferative GN [13]. Diagnosis of C3G is classically made by immunofluorescence on kidney biopsy, with two-fold greater intensity of C3 staining in conjunction with absence or near absence of immunoglobulins [14,15]. C3G comprises of dense deposit disease (DDD), with electron microscopic findings of highly electron-dense, osmiophilic sausage-shaped deposits in the glomerular basement membrane, and C3 glomerulonephritis (C3GN) with deposits in glomerular matrix (subendothelial and few intramembranous) [16,17,18,19,20]. Given the rarity of C3G, it is challenging to derive its precise incidence and prevalence [21]. The prevalence in the United States (US) is estimated at 5 cases per million [21]. Although it is a rare disease, patients with C3G can develop worsening kidney function leading to end-stage kidney disease (ESKD) in up to 50% of patients [22,23,24,25,26,27].

Complement dysregulation is primarily regulated in the fluid phase at the level of C3 convertase, predominantly mediated by genetic mutations and/or triggering factors including infections, monoclonal immunoglobulin, or autoimmune diseases [16,28,29,30,31,32,33,34,35,36,37,38,39,40,41,42,43,44,45,46]. Recent advances in understanding the pathophysiology of complement-mediated diseases have prompted remarkable changes in diagnostic approaches and treatment modalities [47,48,49,50]. Although C3G is a rare glomerular disease, substantially high recurrence rates are noted post kidney transplantation (KTx) [16,50,51,52,53,54,55,56,57,58,59,60,61]. Thus, post-transplant monitoring and appropriate implementation of the available therapies are necessary to improve clinical outcomes [22,50]. Histopathological illustration of C3G recurrence can be detected early post-KTx, with C3G accounting for overall 50% of allograft loss [16,62,63,64]. Among non-KTx patients, multiple treatment modalities have been proposed for C3G, including mycophenolate mofetil, corticosteroids, eculizumab, rituximab, calcineurin inhibitors, cyclophosphamide, and conservative management [16,28,29,30,31,32,33,34,35,36,37,38,39,40,41,42,43,44,45,54,56,65]. However, in KTx patients, despite being on a triple-drug regimen, including mycophenolate mofetil, corticosteroid, and calcineurin inhibitor, C3G still commonly recurs after KTx [16,50]. Currently, the evidence is lacking on the treatment outcomes of C3G among KTx recipients.

Thus, we conducted this systematic review and meta-analysis to assess the efficacy of different treatments for C3G recurrence after KTx.

## 2. Materials and Methods

### 2.1. Data Sources and Search Strategies

The protocol for this systematic review is registered with PROSPERO (International Prospective Register of Systematic Reviews; no. CRD42019125718). A comprehensive search of several databases from each database’s inception to 3 May 2019 was conducted. The databases included OVID MEDLINE (1946 to 3 May 2019), EMBASE (1988 to 3 May 2019), and the Cochrane Database of Systematic Reviews (database inception to 3 May 2019). A systematic literature review was conducted independently by two investigators (M.L.G.S. and C.T.) using the search strategy that consolidated the terms of (“kidney transplantation”, “renal transplantation”, OR “kidney graft”, OR “kidney graft rejection”) AND (“c3 glomerulopathy”, OR “c3 glomerulonephritis”, OR “dense deposit disease”). The actual strategy listing all search terms used is available in the online Appendix A. There were no restrictions on language, sample size, or study duration. This study was conducted by the Preferred Reporting Items for Systematic Reviews and Meta-Analysis (PRISMA) statement [66]. C3G is diagnosed by kidney biopsy with typically dominant C3 staining (at least 2 orders of magnitude above any immunoglobulin deposition) with characteristic deposits (intramembranous sausage-shaped or ribbon-shaped electron-dense deposits in DDD and mesangial, subepithelial, or subendothelial electron-dense deposits in C3GN) as seen by electron microscopy (EM) [2]. Recurrent C3G is diagnosed by biopsy-proven C3G in kidney transplant allograft without prior history of C3G, and de novo C3G is defined by newly diagnosed C3G after kidney transplantation without prior history of C3G [3,67].

### 2.2. Study Selection

Eligible studies must be clinical trials, observational studies (cohort, case-control, or cross-sectional studies), or case series that reported outcomes of adult (age ≥ 18 years old) KTx recipients with C3G. Retrieved articles were individually reviewed for eligibility by the two investigators (M.L.G.S. and C.T.). Discrepancies were addressed and resolved by a third investigator (W.C.). Inclusion was not limited by language, sample size, or study duration.

### 2.3. Data Extraction

The following data were extracted: first author name, year of publication, number of patients, duration of follow-up, type of transplant, mean age, sex, recurrence of C3G after KTx, time from KTx to recurrence, and treatment of G3G. Primary outcome was graft failure.

### 2.4. Data Synthesis and Statistical Analysis

We calculated pooled estimated rates of allograft loss among KTx patients with C3G. Pre-specified subgroup analysis based on the type of C3G (C3GN and DDD) was performed. A random-effects model was used due to the expected clinical heterogeneity in the included populations [68]. All pooled estimates were shown with 95% confidence intervals (CIs). Heterogeneity among effect sizes estimated by individual studies was described with the I^2^ statistic and the chi-square test. A value of I^2^ of 0% to 25% represents insignificant heterogeneity, 26% to 50% low heterogeneity, 51% to 75% moderate heterogeneity and 76 to 100% high heterogeneity [69].

Publication bias was evaluated using the Egger test [70]. A *p*-value of less than 0.05 indicates the presence of publication bias. The meta-analysis was performed by MetaXL software (EpiGear International Pty Ltd., Sunrise Beach, Australia) and the Comprehensive Meta-Analysis 3.3 software (Biostat Inc., Englewood, NJ, USA).

## 3. Result

A total of 207 potentially relevant articles were identified and screened. Twenty-three articles were assessed in detail, of which 12 studies (7 cohort studies and 5 case series) [32,48,50,71,72,73,74,75,76,77,78,79] consisting of 122 KTx patients with C3G (73 C3GN (Table 1) and 49 DDD (Table 2)) were included in our systematic review (Figure 1).

### 3.1. Allograft Loss among KTx Patients with C3G

The pooled estimated rates of allograft loss among KTx patients with C3G were 33% (95% CI: 12–57%) after eculizumab, 42% (95% CI: 2–89%) after therapeutic plasma exchange (TPE), and 81% (95% CI: 50–100%) after rituximab, Figure 2.

### 3.2. Allograft Loss among KTx Patients with C3GN and DDD

A subgroup analysis based on the type of C3G was performed. Pooled estimated rates of allograft loss in C3GN KTx patients were 22% (95% CI: 5–46%) after eculizumab, 56% (95% CI: 6–100%) after TPE, and 70% (95% CI: 24–100%) after rituximab, Figure 3.

Data on allograft loss in DDD KTx patients after different treatment modalities were limited (1 cohort and 1 case series, 4/6 (67%) allograft loss after eculizumab, TPE (1 case series, 0/2 (0%) allograft loss), rituximab (1 cohort, 3/3 (100%) allograft loss), and 2/2 (100%) allograft loss at 6 months after rituximab followed by eculizumab.

Sixty-six patients (38 C3GN, 28 DDD) receive no treatment (due to stable allograft function at presentation and/or clinical judgment of physicians). While there were no statistically significant differences in age, sex, and type of KTx, patients who received treatment for C3G had significant acute kidney injury of kidney allograft and/or proteinuria than those who did not receive treatment (100% vs. 17%, *p* < 0.001). Pooled estimated rates of allograft loss among those who did not receive treatment were 32% (95% CI: 7–64%) and 53% (95% CI: 28–77%) for C3GN and DDD, respectively. Egger’s regression asymmetry test was performed and showed no publication bias (*p* > 0.05 for all analyses).

Among treated C3G patients, data on soluble membrane attack complex of complement (sMAC) were limited to patients treated with eculizumab. 80% patients with elevated sMAC before eculizumab responded to treatment [32,48]. In addition, all patients who responded to eculizumab had normal sMAC levels after post-eculizumab.

## 4. Discussion

KTx patients with C3G who were treated with eculizumab had the lowest rate of allograft loss. The pooled estimated rates of allograft loss were 33% for eculizumab, 42% for TPE and 81% for rituximab. Patients who received no treatment had an estimated graft loss of approximately 40%. The rationale for a clinical response following different treatments could be explained by the underlying pathogenesis of C3G.

In the past, primary MPGN was sub-classified to type I, type II, and type III depending on the location of deposits on electron microscopy [80,81]. Constant efforts in improving the understanding of the complement cascade have led to a shift in classification. Reclassification of MPGN by Sethi et al. as immunoglobulin and non-immunoglobulin mediated disease has paved the pathway to the spectrum of diseases named C3G [15,82,83,84]. C3G is characterized by activation of the alternative complement pathways leading to C3 glomerular deposition [1,2]. With this mechanism, it is suggested that eculizumab may be an effective treatment of C3G [16,62,63,64]. The etiologies of C3G can be classified into genetic causes (mutations/variants resulting in alternative complement pathway abnormalities) and acquired causes (C3Nef/autoantibodies) with possible triggering factors including infections, monoclonal immunoglobulin, and autoimmune diseases [16,26,28,29,30,31,32,33,34,35,36,37,38,39,40,41,42,43,44,45,85,86,87,88].

The availability of robust studies in guiding treatment strategies for recurrent disease post-KTx is lacking. KDIGO recommends the management of C3G based on disease severity [2]. For mild and moderate disease, supportive treatment, along with steroids and mycophenolate, are advised, respectively [2,89]. Rituximab and plasma-exchange have been tried with mixed results [30,90,91,92]. In severe disease with 24 h urine protein >2 g or severe endocapillary proliferation with/without crescent formation, limited success has been described with pulse solumedrol and other immunosuppression therapies [32,33,34,35,36,37,38,39,40,41,42,43,44,45,93]. However, among KTx recipients, despite the immunosuppressive effects of a triple-drug regimen, including mycophenolate, corticosteroid, and calcineurin inhibitor, C3G still recurs after KTx [16,50]. Data on guiding treatment strategies for recurrent C3G post-KTx are lacking. Therefore, the role of anti-complement therapies is being explored.

In this systematic review, we analyzed twelve studies consisting of 122 KTx patients, with the majority carrying a diagnosis of post-KTx C3GN. Living and deceased donors were included in most of the included studies, and the median time from transplantation to the recurrence of the disease varied from as early as 1.5 months to 97 months post-KTx. The Median follow-up period was 6 to 197 months. The pooled estimated rates of allograft loss among KTx patients with C3G were lowest with eculizumab therapy as compared to plasma exchange and rituximab. Subgroup analysis based on the type of C3G revealed similar results. Even with limited data on allograft loss in DDD KTx patients, eculizumab-treated patients demonstrated sustained benefits. Our meta-analysis demonstrated for the first time that KTx patients with C3GN had an associated allograft loss of 22% after treatment with eculizumab, 56% with TPE, and 70% with rituximab. Among 66 patients (38 C3GN, 28 DDD) who received no treatment (due to stable allograft function at presentation and/or clinical judgment of physicians), pooled estimated rates of allograft loss were 32% and 53% among C3GN and DDD groups, respectively. Data on allograft loss in DDD KTx patients who were treated with the above therapies was limited to only one cohort and one case series.

We showed that up to 80% of C3G treated patients with elevated sMAC responded to eculizumab therapy. sMAC levels have been suggested as a serum marker for alternative complement pathway activation [32]. However, the use of sMAC levels to monitor eculizumab treatment in C3G is limited, as the correlation between sMAC level and disease severity has not been established [32,64,94]. Furthermore, several cases have reported C3G KTx patients with normal pre-treatment sMAC levels. Given C3G can develop after KTx despite the use of a triple-drug regimen, including mycophenolate, corticosteroid, and calcineurin inhibitor, the findings from our study suggest that the use of eculizumab for the treatment of C3G after KTx is reasonable. While we found that majority of patients with elevated sMAC prior to treatment responded to eculizumab therapy, future validation studies with a larger number of patients are required.

This study was subject to certain limitations. First, all included studies were observational or case series in design, making them susceptible to selection bias. Second, there is no standard treatment for C3G to allow comparison of interventions. Thus, data from patients who did not receive treatment other than supportive therapy was provided as a reference. Fourth, the rates of remissions or relapses were not reported in most of the included studies. Only the rate of graft loss was available for pooled analysis. The audience should be aware of these limitations when interpreting our findings. Although our meta-analysis suggested that eculizumab might be considered as an additional therapy for C3G in KTx patients, the pooled sample size remains small, and further controlled trials describing the efficacy of eculizumab, TPE, or rituximab are warranted. Lastly, there are currently ongoing clinical trials of complement inhibitors for the treatment of C3G among non-KTx patients, including OMS721 (MASP2 inhibitor: NCT02682407), AMY-101 (C3 inhibitor: NCT03316521), APL-2 (C3 inhibitor: NCT03453619), ACH-4471 (factor D inhibitor: NCT03459443, NCT03369236, and NCT03124368), LNP023 (Factor B inhibitor: NCT03832114), eculizumab (C5 inhibitor: NCT01221181 and NCT02093533), and CCX168 (C5aR1 inhibitor: NCT03301467), respectively [95]. Future studies are required to assess and compare the efficacy and safety of these various complement inhibitors for the treatment of C3G among KTx recipients.

## 5. Conclusions

KTX patients with C3G treated with eculizumab had the lowest incidence of allograft loss (33%) when compared to those treated with TPE and rituximab. Among those who received no treatment for C3G due to stable allograft function, there was an incidence of allograft loss of 32% in C3GN and 53% in DDD.

## Figures and Tables

**Figure 1 medsci-08-00044-f001:**
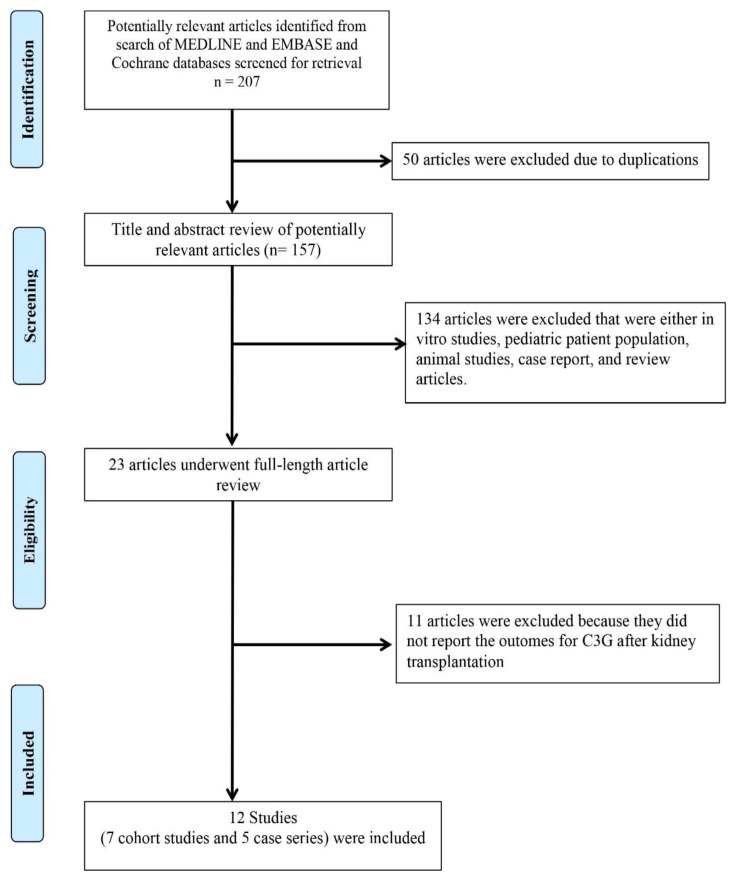
PRISMA flow diagram for study selection.

**Figure 2 medsci-08-00044-f002:**
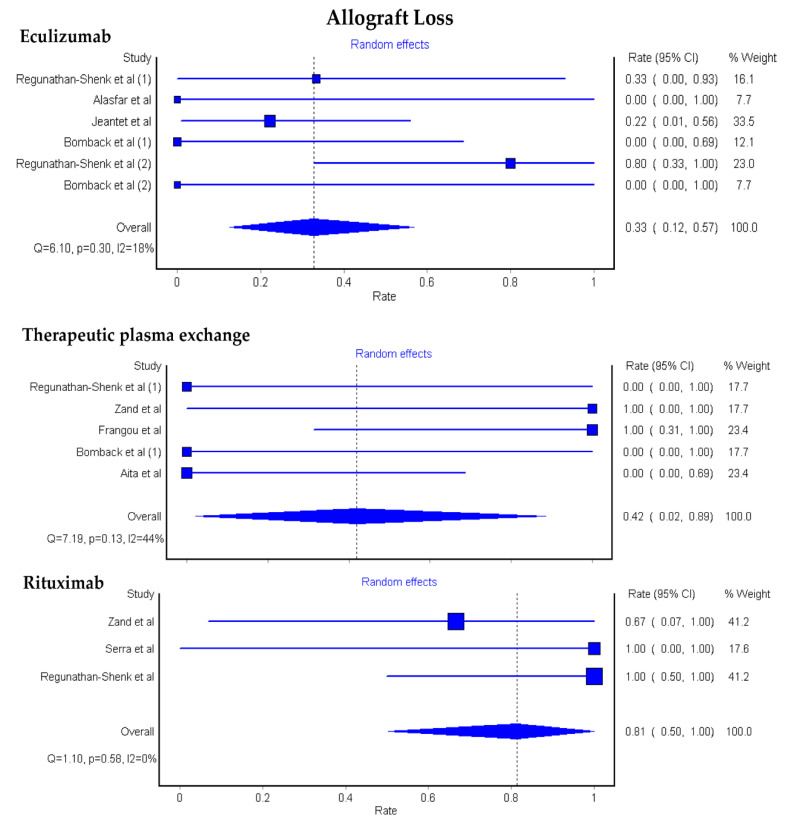
Pooled estimated rates of allograft loss among KTx patients with C3G.

**Figure 3 medsci-08-00044-f003:**
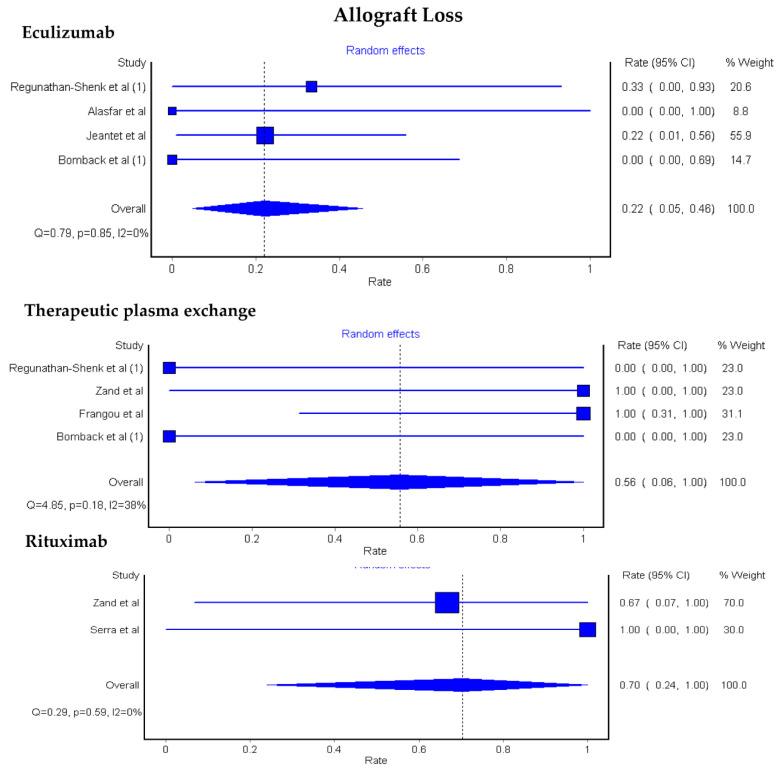
Pooled estimated rates of allograft loss among KTx patients with C3GN.

**Table 1 medsci-08-00044-t001:** Characteristics of the included studies in this systematic review of outcomes of KTx patients with C3GN.

C3GN among KTx Recipients															
Authors	Type of Study	Patients (*n*)	Age at Time of Diagnosis/Transplant, Median (Years)	Females (*n*)	Time to Dialysis or KTX, Median (Months)	Type of KTX	Complement Abnormality	Median Follow-Up (Months)	Median Time from KTx to Recurrence (Months)	Recurrence	Rituximab, Graft Failure (*n*)	Eculizumab, Graft Failure (*n*)	PLEX + Steroids, Graft Failure (*n*)	No Therapy for Recurrence, Graft Failure (*n*)	Graft Failure, Total (*n*)
Regunathan-Shenk et al., 2019 [48]	Cohort	12	22	3	48	LRKTx, 7 LUKTx, 3 DDKTx, 2	CD46, 1 C3Nef, 2 C5Nef, 1 None, 1 Not done, 7	76	-	8, yes 2, probable2, no	0	3, 1 (1 treated with Eculizumab + PLEX)	1, 0	9, 2	3
Zand et al., 2014 [50]	Cohort	21	20.8	9	42.3	LKTx, 17 DDKTx, 4	-	73.9	28	14, yes 7, no	3, 2	0,0	1 + plus autologous peripheral stem cell transplant, 0 1 treated with steroids alone, 1	10, 0	7
Frangou et al., 2019 [71]	Cohort	17	46.7	4	-	LRKTx, 3 LUKTx, 3 DDKTx, 11	CFHR5, 17	157	37	3, yes 9, probable	0,0	0,0	2, 2	14, 3	5
Serra et al., 2018 [72]	Case series	3 (de novo)	66	1	-	DDKTx, 3	None, 2 Anti-CFH ab, 1	-	72	3 de novo	1, 1	0	0	2, 2	3
Wong et al., 2016 [73]	Case series	4 (familial)	26.5	2	-	DDKTx, 4	-	-	97	2, yes	0	0	0	2, 1	1
Alasfar et al., 2016 [74]	Cohort	5	37.4	3	-	DDKTx,1 LUKTx, 1	-	63.6	-	2, yes	0	1,0	0	1, 1	1
Jeantet et al., 2017 [75]	Cohort	9	-	-	-	-	-	-	1.5	9, yes	0	9, 2	0	0	2
Bomback et al., 2012 [32]	Case series	2	21	2	-	-	C3Nef, 2	-	2.5	2	-	2, 0 (1 treated with Eculizumab + PLEX, steroids)	1, 0	0	0

Abbreviations: KTX, kidney transplant; LKTx, living donor kidney transplant, LRKTx, living-related kidney transplant; LUKTx, living unrelated kidney transplant; DDKTx, deceased donor kidney transplant; PLEX, plasma exchange.

**Table 2 medsci-08-00044-t002:** Characteristics of the included studies in this systematic review of outcomes of KTx patients with DDD.

DDD among KTx Recipients															
Authors	Type of Study	Patients, (*n*)	Age at Time of Diagnosis/Transplant, Median (Years)	Females (*n*)	Time to Dialysis or KTx, Median (Months)	Type of KTx	Complement Abnormality	Median Follow-Up (Months)	Median Time from KTx to Recurrence	Recurrence (*n*)	Rituximab, Graft Failure (*n*)	Eculizumab, Graft Failure (*n*)	PLEX, Graft Failure (*n*)	No Therapy, Graft Failure (*n*)	Graft Failure, Total (*n*)
Regunathan-Shenk, 2019 [48]	Cohort	7	30	2	-	LRKTx, 3 LUKTx, 2 DDKTx, 2	C3Nef, 3 CFI, 1 Anti-CFH ab, 1 Not done, 2	-	-	5, true 3, Probable 1, no	3, 3 (1 treated with rituximab + eculizumab) failed (1 treated with rituximab + PLEX), failed (1 treated with rituximab, eculizumab, PLEX), failed	5, 4 (1 treated with eculizumab alone), survived (1 treated with eculizumab alone), failed (1 treated with eculizumab + PLEX) Failed (1 treated with rituximab, eculizumab, PLEX), failed (1 treated with rituximab + eculizumab) failed	3,3 (1 treated with eculizumab + PLEX) Failed (1 treated with rituximab + PLEX), failed (1 treated with rituximab, eculizumab, PLEX), failed	3, 2	7
Aita et al., 2006 [76]	Case series	2	25	0	-	LRKTx, 2	-	6	-	-	0	0	2, 0	0	0
LeQuintrec et al., 2013 [77]	Case series	15	-	-	-	-	-	-	-	5	0	0	0	5, 3	3
Andresdottir et al., 1999 [78]	Cohort	13	23	7	84	DDKTx, 12 LRKTx, 1	-	29	2.9	11	0	0	0	11, 8	8
Droz et al., 1979 [79]	Cohort	11	-	-	-	DDKTx, 7 LRKTx, 4	-	30	4	9	0	0	0	9, 2	2
Bomback et al., 2012 [32]	Case series	1	42	1	-	LRKTx, 1	Negative	-	20	1	0	1, 0	0	0	0

Abbreviations: KTX, kidney transplant; LKTx, living donor kidney transplant, LRKTx, living-related kidney transplant; LUKTx, living unrelated kidney transplant; DDKTx, deceased donor kidney transplant; PLEX, plasma exchange.

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
