# Peer review of "Treatment of C3 Glomerulopathy in Adult Kidney Transplant Recipients: A Systematic Review"

_medsci, 2020, doi:10.3390/medsci8040044_

Round 1
Reviewer 1 Report
I applaud this excellent piece I alluded to a similar phenomenon of response to c3g to complement blockade in my current opinion in Nephrology and hypertension review. I must caution the authors to mention c3 blocking agents as well as c5 blocking agents, as they are now both becoming available. Obviously data for c3G and c3 blockade is still pending.
Author Response
Reviewer 1
I applaud this excellent piece I alluded to a similar phenomenon of response to c3g to complement blockade in my current opinion in Nephrology and hypertension review. I must caution the authors to mention c3 blocking agents as well as c5 blocking agents, as they are now both becoming available. Obviously data for c3G and c3 blockade is still pending.
Response: We thank you for reviewing our manuscript and for your critical evaluation. The reviewer raises important point. We agree with the reviewer. We reviewed all ongoing clinicaltrials in clincaltrials.gov on complement inhibitors for treatment of C3G in our discussion as suggested.
“Lastly, there are currently ongoing clinical trials of complement inhibitors for treatment of C3G among non-KTx patients, including OMS721 (MASP2 inhibitor: NCT02682407), AMY-101 (C3 inhibitor: NCT03316521), APL-2 (C3 inhibitor: NCT03453619), ACH-4471 (Factor D inhibitor: NCT03459443, NCT03369236, and NCT03124368), LNP023 (Factor B inhibitor: NCT03832114), Eculizumab (C5 inhibitor: NCT01221181 and NCT02093533), and CCX168 (C5aR1 inhibitor: NCT03301467), respectively [85]. Future studies are required to assess and compare the efficacy and safety of these various complement inhibitors for treatment of C3G among KTx recipients.”
We greatly appreciated the editors’ time and comments to improve our manuscript. The manuscript has been improved considerably by the suggested revisions.

Reviewer 2 Report
In this Article Dr. Gonzalez-Suarez and co-workers describe a systematic review and meta-analysis aimed to assess the efficacy of different treatments in C3 glomerulopathy (C3G) recurrence/de novo C3G post kidney transplantation (KTx).
The study suggests that “the lowest incidence of allograft loss (33%) among KTx patients with C3G are those treated with eculizumab. Among those who received no treatment for C3G due to stable allograft function, there is a high incidence of allograft loss of 32% in C3 Glomerulonephritis (C3GN) and 53% in Dense Deposit Disease (DDD). sMAC level may help to select good responders to eculizumab.”
The paper addresses an issue of interest.
The following drawbacks are for the Authors’ consideration:
The Authors should describe in more details the criteria that - in the considered articles - allowed the diagnosis of C3G recurrence/C3G de novo in KTx patients.
The classification of C3G (C3GN, DDD) is relatively recent. In order to be able to compare the patients described in the various papers, a review of the diagnostic criteria adopted is essential.
A comparison of the clinical characteristics of treated vs untreated patients is missing.
Was there any data relating to laboratory evaluation (renal function, proteinuria?)
Tables 1 and 2 need to be deeply revised regarding the use of parentheses and miscellaneous errors.
Minor comments:
The choice to consider adult KTx recipients should be made explicit in the Title
We suggest to change in text and Keywords the term “dense deposition disease (DDD)” with the most appropriate “dense deposit disease (DDD)”, also used as search strategy in systematic literature review.
Author Response
Reviewer 2
In this Article Dr. Gonzalez-Suarez and co-workers describe a systematic review and meta-analysis aimed to assess the efficacy of different treatments in C3 glomerulopathy (C3G) recurrence/de novo C3G post kidney transplantation (KTx).
The study suggests that “the lowest incidence of allograft loss (33%) among KTx patients with C3G are those treated with eculizumab. Among those who received no treatment for C3G due to stable allograft function, there is a high incidence of allograft loss of 32% in C3 Glomerulonephritis (C3GN) and 53% in Dense Deposit Disease (DDD). sMAC level may help to select good responders to eculizumab.”
The paper addresses an issue of interest.
Response: We thank you for reviewing our manuscript and for your critical evaluation.
Comment #1. The following drawbacks are for the Authors’ consideration:
The Authors should describe in more details the criteria that - in the considered articles - allowed the diagnosis of C3G recurrence/C3G de novo in KTx patients.
The classification of C3G (C3GN, DDD) is relatively recent. In order to be able to compare the patients described in the various papers, a review of the diagnostic criteria adopted is essential.
Response: We appreciate the reviewer’s input. We agree with the reviewer’s suggestion. We have added the diagnosis criteria or C3G and the definition of C3G recurrence/C3G de novo in KTx patients in the method section of manuscript as the reviewer’s suggestion.
Comment #2.
A comparison of the clinical characteristics of treated vs untreated patients is missing.
Was there any data relating to laboratory evaluation (renal function, proteinuria?)
Response: The reviewer raises important points. We reviewed the included studies again. Although data were limited, we were able to compared available data on treated vs untreated patients. We have additionally added the findings in the result section as reviewer’s suggestion. We reviewed the included studies again, and included studies report data on allograft outcome/graft loss, but data on renal function, proteinuria after the treatment unfortunately were lacking.
“While there were no statistically significant differences in age, sex, and type of KTx, patients who received treatment for C3G had significant acute kidney injury of kidney allograft and/or proteinuria than those who did not receive treatment (100% vs. 17%, p<0.001)."
Comment #3.
Tables 1 and 2 need to be deeply revised regarding the use of parentheses and miscellaneous errors.
Response: We apologize for these errors. We appreciate reviewer’s very thorough reviews. We have corrected the use of parentheses and miscellaneous errors in the table 1 and table 2 as suggested.
Comment #4.
Minor comments:
The choice to consider adult KTx recipients should be made explicit in the Title
Response: We agree with the reviewer. This change has been as suggested.
Comment #5.
We suggest to change in text and Keywords the term “dense deposition disease (DDD)” with the most appropriate “dense deposit disease (DDD)”, also used as search strategy in systematic literature review.
Response: We agree with the reviewer. This change has been as suggested.
We greatly appreciated the editors’ time and comments to improve our manuscript. The manuscript has been improved considerably by the suggested revisions.

Reviewer 3 Report
This is an interesting study. This is very rare to begin with but patients and clinicians face severe challenges when encountered with these conditions especially after kidney transplantation. Hence, this is a relevant study.
The authors acknowledge several limitations of this type of study. These studies should also be interpreted with caution as:
- There are limited number of patients in each series/report
- No standardization of treatment/management options
- No standard methodology as to how each treatment option was chosen or why someone was not treated (other than standard immunosuppression)
As we all know, poor data input from these types of reports will lead to poor data output.
Having said that, there could be some merit to the conclusion that use of Eculizumab might be helpful. It is also amazing that offering no treatment was better than offering TPE or Rituximab! This is where it gets confusing as to the #3 mentioned above.
The authors mention SMAC levels a few times. It would be useful to note which studies mentioned those. Also, what was the average duration of the 3 treatment modalities? Were there any crossover?
This opens the door to more questions. I'm not certain that those could be answered.
It would also be good for the authors to render an opinion as to how they would approach the management of C3G and the limited therapeutic options.
Author Response
Reviewer 3
This is an interesting study. This is very rare to begin with but patients and clinicians face severe challenges when encountered with these conditions especially after kidney transplantation. Hence, this is a relevant study.
Response: We thank you for reviewing our manuscript and for your critical evaluation. We appreciate reviewer’s suggestion.
Comment #1.
The authors acknowledge several limitations of this type of study. These studies should also be interpreted with caution as:
- There are limited number of patients in each series/report
- No standardization of treatment/management options
- No standard methodology as to how each treatment option was chosen or why someone was not treated (other than standard immunosuppression)
As we all know, poor data input from these types of reports will lead to poor data output.
Having said that, there could be some merit to the conclusion that use of Eculizumab might be helpful. It is also amazing that offering no treatment was better than offering TPE or Rituximab! This is where it gets confusing as to the #3 mentioned above.
Response: We appreciate the reviewer’s input. The reviewer raises very important point. We apologize for the confusion. We have thus additionally performed analysis comparing patients who received treatment and those who did not. We have additionally added the findings in the result section as reviewer’s suggestion.
“While there were no statistically significant differences in age, sex, and type of KTx, patients who received treatment for C3G had significant acute kidney injury of kidney allograft and/or proteinuria than those who did not receive treatment (100% vs. 17%, p<0.001)."
In addition, we have additionally clarified this in the conclusion regarding untreated patients. “Among those who received no treatment for C3G due to stable allograft function, there was an incidence of allograft loss of 32% in C3GN and 53% in DDD.”
Comment #2.
The authors mention SMAC levels a few times. It would be useful to note which studies mentioned those. Also, what was the average duration of the 3 treatment modalities? Were there any crossover?
Response: The reviewer raises very important point. We agree and we have additionally cited the studies on sMAC levels the manuscript. We also added information on the crossover as suggested. Only 2 patients with DDD received crossover treatment with both rituximab and eculizumab. We added the additionally findings as suggested.
“2/2 (100%) with DDD had allograft loss at 6 months after rituximab followed by eculizumab.”
Comment #3.
This opens the door to more questions. I'm not certain that those could be answered.
It would also be good for the authors to render an opinion as to how they would approach the management of C3G and the limited therapeutic options.
Response: We appreciate the reviewer’s input. We agree with the reviewer, and we have additionally included more discussion as the reviewer’s suggestion.
“Given C3G can develop after KTx despite the use of triple-drug regimen, including mycophenolate, corticosteroid, and calcineurin inhibitor, the findings from our study suggest that the use of eculizumab for treatment of C3G after KTx is reasonable. While we found that majority of patients with elevated sMAC prior to treatment responded to eculizumab therapy, future validation studies with larger number of patients are required.”
We greatly appreciated the editors’ time and comments to improve our manuscript. The manuscript has been improved considerably by the suggested revisions.

Round 2
Reviewer 2 Report
Line 109-111
Recurrent C3G is diagnosed by biopsy-proven C3G in kidney transplant allograft without prior history of C3G of patient with native C3G, and de novo C3G is defined by newly diagnosed C3G after kidney transplantation without prior history of C3G [3,67]
Reviewer 3 Report
All my concerns/comments have been addressed. Thanks.